# Major transcriptional changes observed in the Fulani, an ethnic group less susceptible to malaria

Jaclyn E Quin[1‡], Ioana Bujila[1‡], Mariama Chérif[2,3‡], Guillaume S Sanou[2‡], Ying Qu[4], Manijeh Vafa Homann[5†], Anna Rolicka[1§], Sodiomon B Sirima[2], Mary A O'Connell[6], Andreas Lennartsson[4], Marita Troye-Blomberg[1], Issa Nebie[2], Ann-Kristin Östlund Farrants[1*]

[1]Department of Molecular Biosciences, The Wenner-Gren Institute, Stockholm University, Stockholm, Sweden; [2]Centre National de Recherche et de Formation sur le Paludisme, Ouagadougou, Burkina Faso; [3]Université Polytechnique de Bobo-Dioulasso, Bobo-Dioulasso, Burkina Faso; [4]Department of Biosciences and Nutrition, Karolinska Institute, Stockholm, Sweden; [5]Unit of Infectious Diseases, Department of Medicine, Karolinska Institute, Stockholm, Sweden; [6]Central European Institute of Technology, Brno, Czech Republic

*For correspondence:
anki.ostlund@su.se

[†]These authors contributed equally to this work
[‡]These authors also contributed equally to this work

Present address: [§]Faculty of Animal Sciences Department of Genetics and Animal Breeding, SGGW, Warsaw University of Life Sciences, Warszawa, Poland

Competing interests: The authors declare that no competing interests exist.

**Abstract** The Fulani ethnic group has relatively better protection from *Plasmodium falciparum* malaria, as reflected by fewer symptomatic cases of malaria, lower infection rates, and lower parasite densities compared to sympatric ethnic groups. However, the basis for this lower susceptibility to malaria by the Fulani is unknown. The incidence of classic malaria resistance genes are lower in the Fulani than in other sympatric ethnic populations, and targeted SNP analyses of other candidate genes involved in the immune response to malaria have not been able to account for the observed difference in the Fulani susceptibility to *P.falciparum.* Therefore, we have performed a pilot study to examine global transcription and DNA methylation patterns in specific immune cell populations in the Fulani to elucidate the mechanisms that confer the lower susceptibility to *P.falciparum* malaria. When we compared uninfected and infected Fulani individuals, in contrast to uninfected and infected individuals from the sympatric ethnic group Mossi, we observed a key difference: a strong transcriptional response was only detected in the monocyte fraction of the Fulani, where over 1000 genes were significantly differentially expressed upon *P.falciparum* infection.
DOI: https://doi.org/10.7554/eLife.29156.001

## Introduction

The Fulani ethnic group has relatively better protection from *Plasmodium falciparum* malaria than other ethnic groups living alongside them in West Africa. Since the first report of the different response of Fulani to *P.falciparum* in 1996 (**Modiano et al., 1996**), populations of Fulani living sympatrically with other ethnic groups from Mali to as far east as Sudan, have consistently been reported to have fewer symptomatic cases of malaria, lower *P.falciparum* infection rates, and lower *P.falciparum* density in infected individuals (**Arama et al., 2015a**). However, the basis for the lower susceptibility of Fulani to malaria is yet to be established.

Malaria has been a major selective force on the human genome in endemic areas, and a number of factors conferring reduced susceptibility to malaria have been described, such as enzymatic G6PD deficiency, or abnormal hemoglobins - the most well know of which, HbS, confers sickle cell disease (**Allison, 2009**). However, the incidence of these classic malaria resistance genes are lower in

**eLife digest** There is a huge international effort to combat malaria, but even today almost half a million people die from the disease each year, mostly young children in Africa. Malaria infections are caused by the parasite *Plasmodium*, which is spread by mosquitoes. The Fulani are an ethnic group of people from West Africa that are naturally better at fighting malaria infections with the most dangerous form of the parasite, *Plasmodium falciparum*. Fulani show fewer symptoms of malaria and carry fewer parasites when they are infected. They also have fewer cases of sickle cell disease, a condition that is known to protect against malaria. Yet, no one understands what it is that makes Fulani more resistant to malaria than other people, such as people of other ethnic groups that live in the same region of West Africa but who do not intermarry with the Fulani.

Past studies that looked at likely genes, such as those involved in the immune response, could not find any differences between the Fulani and people who have a normal susceptibility to malaria. Quin, Bujila et al. performed a pilot study to look at the activity of all the genes in immune cells from Fulani people who had become naturally infected with *P. falciparum* to see which genes are switched on or off after an infection. If some genes are used differently in the Fulani compared to other ethnic groups, then it is likely that these genes are responsible for the Fulani's more effective immune response to *P. falciparum.*

First, Quin, Bujila et al. looked for chemical markers that are naturally added to DNA to influence the activity of nearby genes, and used other methods to determine which genes are switched on and at what level. Unexpectedly, for those markers looked at, no difference was found between the ethnic groups investigated. However, the other experiments did show that in a certain type of immune cell in the Fulani over 1,000 genes become notably more active, or less active, after *P. falciparum* infection. These cells, called monocytes, are important for the immune system's first line of defence, which begins the attack against an infection and alerts the rest of the immune system. Lastly, inflammation is a common part of the body's immune response to many parasites. When Quin, Bujila et al. measured the level of inflammatory molecules, they found that Fulani have higher levels of molecules that promote inflammation than other ethnic groups.

Together these results suggest that a group of genes in the monocytes of Fulani are set in a 'primed' state, which helps monocytes to drive the fight against *P. falciparum* more effectively. The cause for the heightened primed state remains unclear, but previous studies looking into bacterial and fungal infections have shown early infections can prime the immune system to promote more inflammation during a second infection. These new findings suggest that such processes might also occur for malaria infections, and so might represent a new avenue of research in the quest for better treatments for malaria.

DOI: https://doi.org/10.7554/eLife.29156.002

the Fulani than in other sympatric ethnic populations (*Modiano et al., 2001*; *Mangano et al., 2015*), and therefore cannot account for their better protection. Neither have targeted SNP analysis of candidate genes involved in the immune response to malaria been able to explain the observed difference in Fulani susceptibility to *P.falciparum* (*Arama et al., 2015a*). Studies examining specific characteristics of the immune response to malaria in Fulani have shown a number of differences – Fulani are more responsive to *P.falciparum* antigens, with higher levels of *P.falciparum* specific IgG1-3, IgE and IgM antibodies (*Modiano et al., 1996*; *Arama et al., 2015a*; *Arama et al., 2015b*; *Bolad et al., 2005*; *Vafa et al., 2009*; *Boström et al., 2012*; *Cherif et al., 2012*; *Maiga et al., 2014*; *Cherif et al., 2015*; *Cherif et al., 2016*); Fulani have a higher ratio of pro-inflammatory to anti-inflammatory cytokines (*Boström et al., 2012*; *McCall et al., 2010*; *Arama et al., 2011*; *Farouk et al., 2005*); Fulani have a different proportion of specific immune cells, including more activated memory B-cells (*Portugal et al., 2012*), fewer circulating regulatory T-cells (*McCall et al., 2010*; *Torcia et al., 2008*), and increased activation of dendritic cells correlating with their lower frequency in circulating blood (*Arama et al., 2011*). However, no direct associations have been established between these differences and reduced parasite rates in Fulani individuals. Thus, the contribution of differences in immune response in the Fulani to protection from malaria is unclear, as is their underlying cause.

To shed light on the mechanisms that confer the lower susceptibility of Fulani to *P.falciparum* malaria, we have performed a pilot study examining global DNA methylation and transcription regulation in the Fulani, compared to the Mossi sympatric ethnic group. We have examined both CD14+ (monocyte) and CD14- (predominantly lymphocyte) populations of peripheral blood mononuclear cells from the same individuals, either uninfected or infected with *P.falciparum*. Our results show that Fulani monocytes, specifically, are more transcriptionally reactive to *P.falciparum* infection. This is not related to differences in DNA-methylation. Rather, several genes involved in chromatin remodelling and epigenetic regulation of gene expression in immune cell lineages are differentially expressed, indicating that the underlying cause is a change in the chromatin landscape. The heightened transcriptional response is associated with differential expression of a variety of genes involved in the innate immune response, and corresponds with more inflammatory baseline characteristics, and a more activated inflammasome upon *P.falciparum* infection, in Fulani individuals.

## Results

This is a pilot study of adolescent men from Bakoundouba (Burkina Faso), where the Fulani live in sympatry with the Mossi ethnic group (*Modiano et al., 1996*; *Lulli et al., 2009*). Due to the 2014 West African Ebola outbreak and local political unrest, the collection was delayed until the end of the July-October high transmission season, to November 2014. All individuals (56) were analysed for *P.falciparum* infection by PCR, in which other *Plasmodium* infections were also detected. The baseline characteristics presented in *Figure 1a* show that the Fulani population had a significantly lower rate of *P.falciparum* infection (*Figure 1b*), and significantly higher levels of *P.falciparum* specific antibodies (IgG) (*Figure 1c*), consistent with previous reports (*Boström et al., 2012*; *Arama et al., 2011*; *Perdijk et al., 2013*). After having excluded samples with mixed *Plasmodium* infections, G6PD-deficiency, or hemoglobin C, samples of CD14$^+$ (monocytes) and CD14$^-$ (~98% lymphocytes) cells from infected and uninfected Fulani and Mossi individuals were selected for RNA-sequencing and DNA-methylation analyses (*Figure 1—figure supplement 1*). Samples were selected from the same individuals for each of the the different analyses. Samples from the remaining individuals formed a separate cohort for validation experiments.

RNA-sequencing analysis of monocytes from uninfected Fulani (n = 7) compared to uninfected Mossi (n = 4) identified 17 significantly differentially expressed (DE) genes (*Figure 2a*). Analysis by qRT-PCR of a subset of these genes in the validation cohort revealed similar changes in expression, indicating that our results reflected the larger Fulani population (*Figure 2b*). The most significantly DE gene was *P2RX 7* (*Figure 2c*), encoding P2X purinoceptor 7, a purinergic receptor for ATP that acts as a ligand-gated ion channel (*Wiley et al., 2011*). P2RX7 is highly expressed in immune cells, particularly monocytes, and its activation drives the formation of the inflammasome and subsequent secretion of the pro-inflammatory cytokines IL-1β and IL-18 (*Sharma and Kanneganti, 2016*). DNA methylation analysis of monocytes from the Mossi and the Fulani revealed only a few sites that were significantly differentially methylated (*Supplementary file 1A*). These sites did not overlap with the DE genes identified by RNA-sequencing. No DE genes were identified in the CD14$^-$ cells from uninfected Fulani and Mossi.

We next analysed the transcriptional response of monocytes to *P.falciparum* infection (*Supplementary file 2A* and *Figure 2—figure supplements 1* and *2*). A striking difference between Fulani and Mossi was observed in infected compared to uninfected monocytes: infected Fulani had 1239 DE genes, most of which were downregulated (1095/1239) (*Figure 2d and f*, *Supplementary file 2B*); in comparison, infected Mossi had only three DE genes (*Figure 2e and f*, *Supplementary file 2C*). The CD14$^-$ fraction from the same individuals did not display differences in gene expression (*Figure 2g and h*).

The most significantly DE genes in monocytes of infected Fulani are involved in a variety of processes, including immune response, metabolism, cell cycle, chromatin and transcription regulation, RNA metabolism, translation, and ribosomal biogenesis (*Figure 3a*). Next we examined whether the response was quantitatively (ie different changes in level of expression of the same gene sets) or qualitatively (ie changes in the level of expression of different gene sets) between Fulani and Mossi following infection. Only a few genes were significantly differentially expressed between the two infected populations, suggesting that the majority of significantly DE genes in Fulani also experienced less significant changes in expression in Mossi. (*Figure 3—figure supplement 1* and

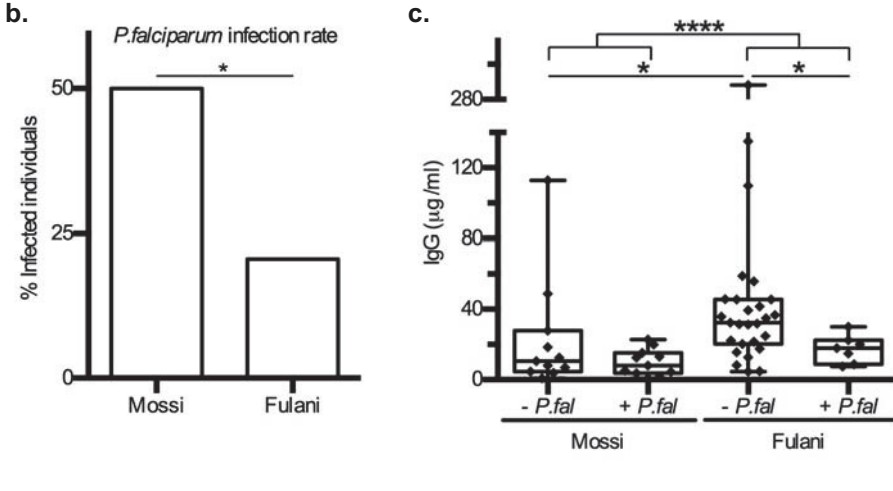

**a. Description of study population.**

| n=56 | Fulani (n=34) | Mossi (n=22) | p-value |
|---|---|---|---|
| Mean age | 18.97 (15-24) | 17.1 (15-22) | **p=0.008[a] |
| *P.falciparum* infection | 20.6% (n=7) | 50.0% (n=11) | *p=0.0388[b] |
| Mean *Plasmodium* density ( /µl blood) | 631.0 (n=6) | 833.2 (n=5) | n.s. (p=0.7922)[a] |
| Mean *P.falciparum* specific IgG (µg/ml) | 40.41 | 16.51 | ****p<0.0001[a] |
| Mean axillary temp. (C) | 36.42 (36-37.1) | 36.49 (36-37.4) | n.s.[a] |
| Mean Hb level (g/dl) | 13.18 (8.1-15.7) | 12.93 (10-16.1) | n.s.[a] |
| G6PD status | 14.7% (n=5) | 4.5% (n=1) | n.s. (p=0.3862)[b] |
| HB type | 79.4% AA (n=27) | 77.3% AA (n=17) | |
| | 20.6% AC (n=7) | 18.2% AC (n=4) | |

[a] two-tailed Mann-Whitney non-parametric U test; [b] two-tailed Fisher's exact test.

**Figure 1.** Characteristics of *P.falciparum* infection in Mossi and Fulani. (a) Table describing study population. (b) Rate of infection with *P.falciparum* (Mossi n = 11/22 and Fulani n = 7/34, *p=0.0388). (c) Plasma levels of IgG antibodies to *P.falciparum* schizont extract antigens. Uninfected (- *P.fal*) and Infected (+*P.fal*). n = 11 Mossi – *P.fal*[w], 11 Mossi +*P.fal*[x], 27 Fulani – *P.fal*[y], 7 Fulani +*P.fal*[z] (median ±min to max; w vs x n.s. p=0.4385, y vs z *p=0.0189, w vs y *p=0.0141, x vs z n.s. p=0.0693, w and x vs y and z ****p<0.001).

DOI: https://doi.org/10.7554/eLife.29156.003

The following figure supplement is available for figure 1:

**Figure supplement 1.** Schematic representation of workflow.

DOI: https://doi.org/10.7554/eLife.29156.004

*Supplementary file 2D*). This indicates that Fulani monocytes have a stronger transcriptional response.

The DE genes related to immune response include NF-κB regulation, MAPK regulation, AP-1 regulation, inflammasome activation, glytolytic metabolism, migration and vesicle function, and other genes involved in innate immune responses (*Figure 3a*). Analysis by qRT-PCR of the expression of a subset of genes in monocytes of individuals from the validation cohort showed a similar downregulation in expression as observed in the RNA-sequencing data (*Figure 3b and c*, *Figure 3—source data 1*). The qRT-PCR results also indicate that these transcript were present at higher levels in monocytes from uninfected Fulani than Mossi (*Figure 3d*). Thus, Fulani monocytes have a more transcriptionally active baseline, and express these genes at relatively high levels prior to infection with *P.falciparum*.

Previous studies have reported that the Fulani have a higher ratio of pro-inflammatory to anti-inflammatory cytokines (*Boström et al., 2012*; *McCall et al., 2010*; *Torcia et al., 2008*; *Perdijk et al., 2013*), and in agreement with this the levels of the pro-inflammatory IFN-γ were higher, and the levels of the anti-inflammatory TGF-β were lower, in blood plasma from uninfected Fulani than uninfected Mossi in our study (*Figure 3e and f*). However, in individuals that were

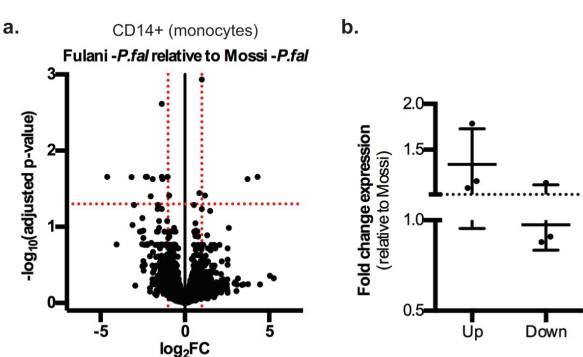

**c. Table of significantly DE genes in CD14+ monocytes of –*P.fal* Fulani relative to –*P.fal* Mossi** (adjusted p-value < 0.05).

| Gene symbol[a] | Description | log$_2$FC | Adjusted p-value[b] |
|---|---|---|---|
| **Fulani –*P.fal* relative to Mossi –*P.fal*** | | | |
| P2RX7 | Purinergic receptor P2X, ligand gated ion channel, 7 | **1.00** | 0.0012 |
| C15orf48 | Chromosome 15 open reading frame 48 | -1.36 | 0.0024 |
| DACH1 | Dachshund family transcription factor 1 | -1.37 | 0.0221 |
| GPRIN3 | GPRIN family member 3 | -1.02 | 0.0221 |
| ADAM28 | ADAM metallopeptidase domain 28 | -1.04 | 0.0221 |
| ADGRG3 | Adhesion G protein-coupled receptor G3 | -2.25 | 0.0221 |
| ANXA3 | Annexin A3 | -2.31 | 0.0221 |
| ENSG00000268240 | Uncharacterized ncRNA | -4.60 | 0.0221 |
| RNF150 | Ring finger protein 150 | **4.31** | 0.0221 |
| PI3 | Peptidase inhibitor 3 | -3.16 | 0.0222 |
| IQCJ-SCHIP1 | IQCJ-SCHIP1 readthrough | -1.33 | 0.0237 |
| BEX1 | Brain expressed, X-linked 1 | **3.72** | 0.0237 |
| FGF13 | Fibroblast growth factor 13 | -1.91 | 0.0237 |
| CPVL | Carboxypeptidase, vitellogenic-like | **0.86** | 0.0361 |
| CD82 | CD82 molecule | -0.94 | 0.0389 |
| FAM26F | Family with sequence similarity 26, F | **1.19** | 0.0389 |
| EPHB1 | EPH receptor B1 | -2.03 | 0.0398 |

[a] Where no gene symbol is assigned, Ensembl gene identification numbers are given. [b] adjusted p-values control for false discovery rate (FDR) using the Benjamini-Hochberg procedure.

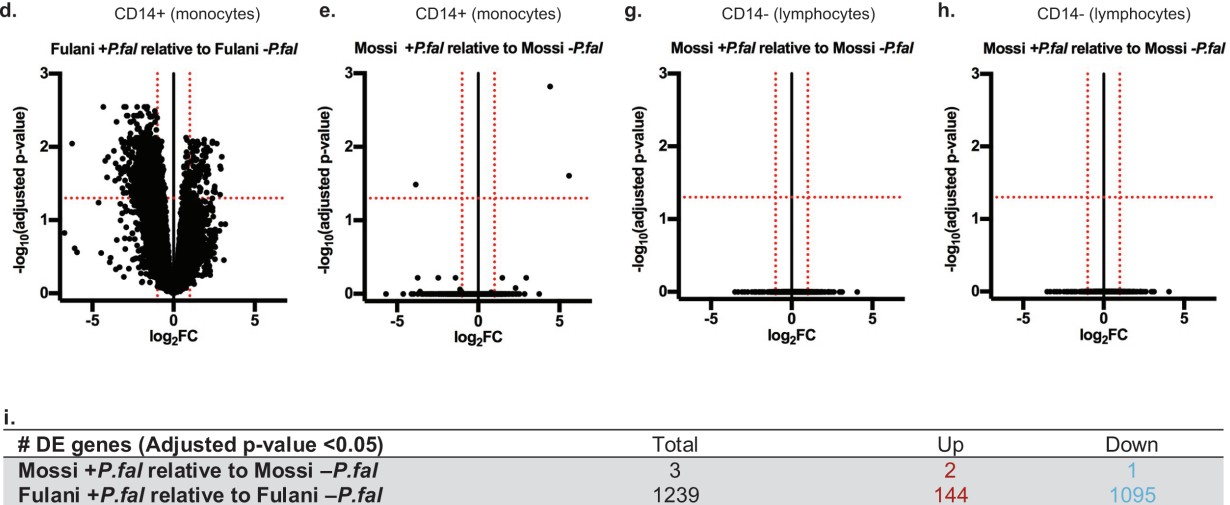

| # DE genes (Adjusted p-value <0.05) | Total | Up | Down |
|---|---|---|---|
| **Mossi +*P.fal* relative to Mossi –*P.fal*** | 3 | 2 | 1 |
| **Fulani +*P.fal* relative to Fulani –*P.fal*** | 1239 | 144 | 1095 |

**Figure 2.** Differential gene expression in Mossi and Fulani following *P.falciparum* infection. (**a**) Volcano plot of expression of individual genes in CD14 + cells (monocytes) of Fulani –*P.fal* (n = 7) relative to Mossi –*P.fal* (n = 4). (**b**) qRT-PCR analysis of selected DE genes (Up: *P2RX7, CPVL, FAM26F*. Down: *ADAM28, ADGRG3, EPBH1*) in validation cohort Mossi –*P.fal* (n = 4) and Fulani –*P.fal* (n = 10) (mean ±SD). (**c**) Table of significantly DE genes in CD14 + cells (monocytes) of Fulani –*P.fal* (n = 7) relative to Mossi –*P.fal* (n = 4). (**d,e**) Volcano plot of expression of individual genes in CD14+ cells

*Figure 2 continued on next page*

Figure 2 continued

(monocytes) of Fulani +*P.fal* (n = 2) relative to Fulani -*P.fal* (n = 7) (d), and Mossi +*P.fal* (n = 3) relative to Mossi -*P.fal* (n = 4) (e). (f) Table of number of DE genes in CD14+ cells (monocytes) of Mossi +*P.fal* relative to Mossi –*P.fal* and Fulani +*P.fal* relative to Fulani –*P.fal*. (g,h) Volcano plot of expression of individual genes in CD14- cells (lymphocytes) of Fulani +*P.fal* (n = 3) relative to Fulani -*P.fal* (n = 5) (g), and Mossi +*P.fal* (n = 3) relative to Mossi –*P.fal* (n = 5) (h).

DOI: https://doi.org/10.7554/eLife.29156.005

The following figure supplements are available for figure 2:

**Figure supplement 1.** Heat map of Pearson correlation between FPKM (fragments per kilobase of transcript per million mapped reads) values of for all CD14+ (monocytes) samples.

DOI: https://doi.org/10.7554/eLife.29156.006

**Figure supplement 2.** Multidimension scaling for all CD14+ (monocytes) samples.

DOI: https://doi.org/10.7554/eLife.29156.007

infected with *P.falciparum*, the levels of pro-inflammatory IFNγ and IL-6 were increased in the Mossi but reduced in the Fulani (*Figure 3f*). While studies have shown higher overall levels of IFNγ in Fulani (*Boström et al., 2012*; *McCall et al., 2010*; *Torcia et al., 2008*), the reduction of IFNγ or other pro-inflammatory cytokines in Fulani upon infection has not been previously reported. Interestingly, the changes in the levels of these cytokines mirrors the expression of the immune response genes above. Collectively, these results indicate monocytes of uninfected Fulani are more immune reactive, driving different dynamics of response upon *P.falciparum* infection.

Many of the DE genes in monocytes from infected Fulani are involved in the function of the inflammasome (*Supplementary file 2E*). The inflammasome is a cytoplasmic complex responsible for processing and activation of pro-inflammatory cytokines IL-1β and IL-18, and causing an inflammatory form of cell death - pyroptosis - via processing and activation of gasdermin D (*Guo et al., 2015*). Monocytes from infected Fulani had significantly reduced expression of inflammasome pathway components (for example the gene encoding gasdermin D, *GSDMD,* and inflammasome complex scaffold proteins, *NLRP12* [*Ataide et al., 2014*] and *NLRC5* [*Davis et al., 2011*]). Therefore, we investigated the activation of the inflammasome by analyzing levels of its targets, IL-1β and IL-18, in blood plasma. IL-1β was only detected in infected Fulani, and IL-18 was detected at higher levels in infected than uninfected Fulani, whereas the levels in the Mossi did not change (*Figure 3g*). This indicates that the inflammasome was significantly more activated in infected Fulani than in the Mossi. As the levels of IL-1β and IL-18 are relatively low in blood plasma, we examined the levels of suppressors of inflammasome signaling, including IL-1 receptor antagonist (IL-1Ra), the soluble decoy IL-1 receptor, type II (IL-1RII), and decoy receptor IL-18 binding protein (IL-18BP), which have effective concentrations several orders of magnitude higher their target cytokines (*Boraschi and Tagliabue, 2013*). These showed similar results, with higher levels in infected Fulani (*Figure 3—figure supplement 2*; IL-1Ra **p=0.0048, and IL-18BPa *p=0.0149, in infected compared to uninfected Fulani). Interestingly, this is in contrast to levels of other pro-inflammatory cytokines, IFNγ and IL-6, which are higher in uninfected than infected Fulani. Our results suggest that the Fulani respond with a more active inflammasome, which results in feedback inhibition of the expression of inflammasome pathway components.

Analysis of transcription factor binding sites at the most significantly DE genes showed enrichment for ETS family transcription factors, which are downstream regulators of the MAPK and PI3K signalling pathways (*Yordy and Muise-Helmericks, 2000*). A number of MAPK and PI3K pathway components were differentially expressed, as was the ETS family transcription factor ERG. Significantly DE genes were also enriched in transcription factor binding sites for NF-κB and NRF2, signalling pathways that are directly regulated in activation of innate immune response pathways in monocytes (*Dev et al., 2011*; *Kobayashi et al., 2013*). A number of NF-κB regulatory components, as well as the NRF2 binding partner MAFK, were also differentially expressed. Many of these transcription factors are associated with coactivators, responsible for histone modification and altered chromatin accessibility (*Martens et al., 2012*; *Saeed et al., 2012*) (*Supplementary file 2F*).

In addition to genes directly associated with innate immune response, genes involved in inducing different chromatin states and RNA metabolism were differentially expressed in monocytes from infected Fulani (*Figure 4a*). This suggests that monocytes from the Fulani have a capacity to remodel the chromatin landscape, which could result in a different or stronger transcriptional responses.

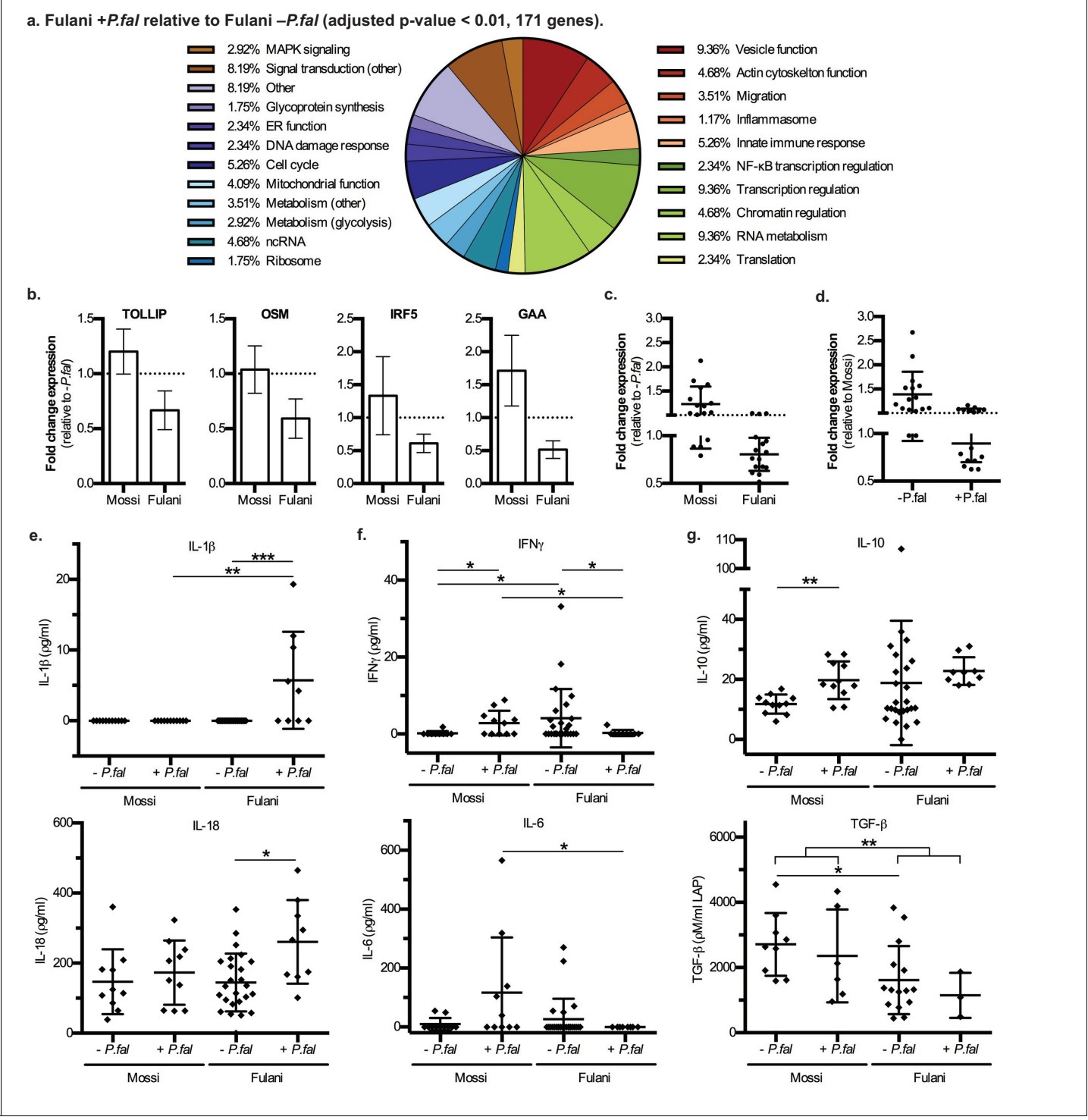

**Figure 3.** Characteristics and validation of significantly DE genes in CD14+ cells (monocytes) of Fulani+*P.fal* relative to Fulani -*P.fal*. (**a**) Classification of the most significantly DE genes in CD14+ cells (monocytes) of Fulani +*P.fal* relative to Fulani –*P.fal* (adjusted p-value<0.01, 171 genes). (**b,c,d**) qRT-PCR analysis of selected DE genes, with individual genes relative to –*P.fal* individuals (**b**), and combined genes relative to –*P.fal* (**c**) or Mossi (**d**) individuals (Genes: *ALDOA, ARID3A, DUSP8, FES, FURIN, GAA, GAPDH, IRF3, IRF5, NFKBIE, OSM, RFXANK, SQSTM1, STRA13, TOLLIP, TRAF7*), in validation cohort of Mossi –*P.fal* (n = 4), Fulani –*P.fal* (n = 10), Mossi +*P.fal* (n = 8) and Fulani + *P.fal* (n = 5) (mean ±SD) (Also see *Figure 3—source data 1*). (**e,f,g**) Cytokine levels in blood plasma of Mossi and Fulani individuals following *P.falciparum* infection. Uninfected (-*P.fal*) and Infected (+*P.fal*). n = 11 Mossi – *P.fal*[w], 11 Mossi +*P.fal*[x], 25 Fulani –*P.fal*[y], 9 Fulani + *P.fal*[z](**e**) Inflammasome cytokines IL-1β (mean ±SD; w vs x n.s. p>0.9999, y vs z ***p=0.0005, w vs y n. s. p=0.9285, x vs z **p=0.0081) and IL-18 (mean ±SD; w vs x n.s. p=0.4813, y vs z *p=0.0106, w vs y n.s. p>0.9999, x vs z n.s p=0.0947). (**f**) Pro-

*Figure 3 continued on next page*

*Figure 3 continued*

inflammatory cytokines IFNγ (mean ±SD; w vs x *p=0.0456, y vs z *p=0.0343, w vs y *p=0.0409, x vs z *p=0.0243) and IL-6 (mean ±SD; w vs x n.s. p=0.0862, y vs z n.s. p=0.3023, w vs y n.s. p=0.7166, x vs z *p=0.0359) (g) TGFβ (mean ±SD; w vs x n.s. p=0.5287, y vs z n.s. p=0.5735, w vs y *p=0.0122, x vs z n.s. p=0.2619, w and x vs y and z **p=0.0059) and IL-10 (mean ±SD; w vs x **p=0.032, y vs z n.s. p=0.0659, w vs y n.s. p=0.7154, x vs z n.s. p=0.2014)

DOI: https://doi.org/10.7554/eLife.29156.008

The following source data and figure supplements are available for figure 3:

**Source data 1.** qRT-PCR analysis of selected DE genes in monocytes of validation cohort of Mossi –*P.fal* (n = 4), Fulani –*P.fal* (n = 10), Mossi +*P.fal* (n = 8) and Fulani +*P.fal* (n = 5) (mean ±SD).
DOI: https://doi.org/10.7554/eLife.29156.011

**Figure supplement 1.** Volcano plot of expression of individual genes in Fulani +*P.fal* (n = 2) relative to Mossi +*P.fal* (n = 3) CD14 +PBMCs (monocytes).
DOI: https://doi.org/10.7554/eLife.29156.009

**Figure supplement 2.** ELISA of blood plasma in Mossi and Fulani individuals following P.falciparum infection.
DOI: https://doi.org/10.7554/eLife.29156.010

Several important genes encoding for chromatin and epigenetic factors are differentially regulated. For example PRDM5, which is expressed in heamatapoietic stem cells where it is predicted to regulate chromatin organisation via association with architectural proteins CTCF, Cohesin, and TFIIIC (*Riddell et al., 2014*; *Galli et al., 2013*), was upregulated in monocytes of infected Fulani. Several chromatin remodelling components and histone modifying enzymes related to a more open chromatin state were differentially expressed; in particular components of the the SWI/SNF complexes, which are important for transcription of many immune genes and the phase shift to the adaptation phase in B-cells and T-cells (*Foster et al., 2007*; *Schultze, 2017*), and components of histone methyl transferase complexes SET1 and MLL complexes, both H3K4-me3 methyl transferases important in haematopoiesis (*Yang and Ernst, 2017*). Genes encoding enzymes responsible for inducing a silent state, such as SUV39H1, SUV420H1 and HDACs, were also differentially expressed, indicating that the chromatin response to *P.falciparum* is complex. The downregulation of many of these factors in infected Fulani were confirmed in the validation cohort *Figure 4b and c*, *Figure 4—source data 1*). Similar to the immune genes above, genes that are upregulated in response to *P.falciparum* infection in Mossi, are already expressed at higher levels in uninfected Fulani. The observed differences in expression in monocytes of the Fulani was not caused by DNA-methylation; the DNA methylation was very stable, with few differences between all analysed sample groups (*Figure 4—figure supplement 1*). The few genes differentially methylated in monocytes from infected and uninfected Fulani did not correspond to the DE genes (*Figure 4d*, *Supplementary files 1B*). Further analysis of DE genes in monocytes of infected Fulani, examining their chromosomal position, showed that they cluster in regions (*Figure 4e*). These regions displayed few differences in DNA methylation (*Figure 4g*). This suggests that a changed chromatin landscape underlies the altered gene expression pattern.

## Discussion

In this pilot study, we have established that monocytes from the Fulani are strikingly more transcriptionally reactive than a sympatric ethnic group in response to infection with *P.falciparum*. Interestingly, most genes exhibited a lower expression level in infected compared to uninfected monocytes. We speculate that this, seen at the end of the malaria transmission season, is due to one, or both, of the following scenarios: First, the higher levels of transcription of immune response genes in uninfected Fulani may drive a more immediate response following *P.falciparum* infection, and thus a more rapid and evident return to homeostatic levels of transcription; Second, an acute hyper-active response following *P.falciparum* infection may result in an subsequent hypo-active state. This is supported by our results showing that genes involved in pathways which are rapidly activated under inflammatory conditions are significantly downregulated in infected Fulani, for example genes encoding aerobic glycolytic metabolism proteins (GAA and GAPDH; *Figure 3—source data 1*) (*Lachmandas et al., 2016*; *McCall et al., 2011*).

Changes in transcription in monocytes of *P.falciparum* infected Fulani were associated with a differential regulation of innate immune response pathways, in particular, the activation of the

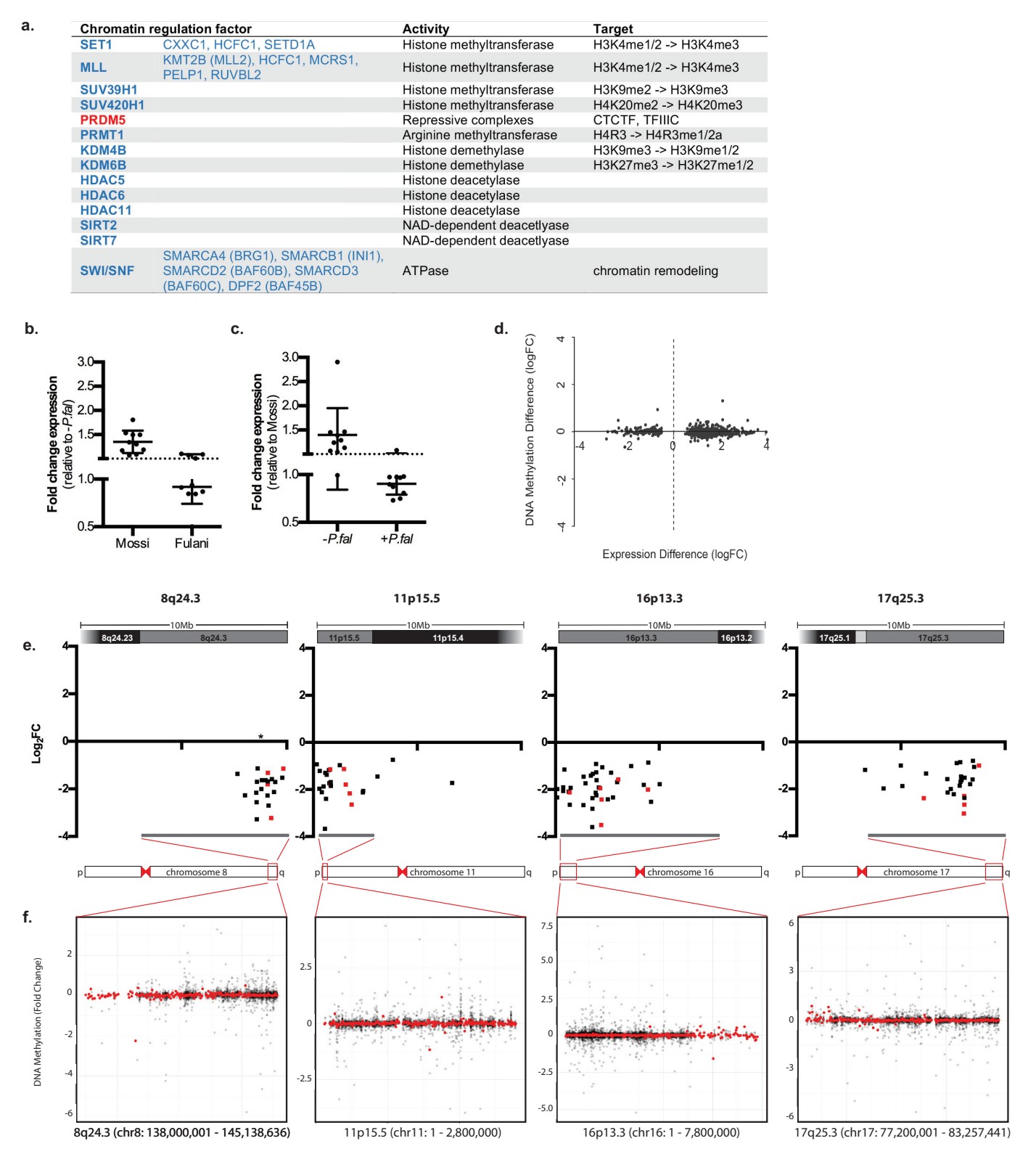

**Figure 4.** Regulation of chromatin in Mossi and Fulani monocytes. (a) Table of selected DE chromatin regulation factors. (b and c) qRT-PCR analysis of selected DE chromatin regulation genes (Genes: *SETD1A, CXXC1, KDM4B, KDM6B, BRG1, RUVBL2, SUV39H1, SIRT7, PHC2, PPP4C*) in validation cohort of Mossi -*P.fal* (n = 4), Fulani -*P.fal* (n = 10), Mossi +*P.fal* (n = 8) and Fulani +*P.fal* (n = 5) relative to –*P.fal* (b) and Mossi (c) individuals (mean ±SD). (Also see *Figure 4—source data 1*). (d) Difference in DNA methylation at DE genes (Fulani +*P.fal* (n = 3) relative to Fulani –*P.fal* (n = 3)). (e and f)
*Figure 4 continued on next page*

*Figure 4 continued*

Examples of clusters of DE genes. (e) Change in expression of all significantly DE genes, in selected regions that include cytobands enriched in DE genes (Fulani +*P.fal* relative to Fulani −*P.fal*, adjusted p-valued < 0.05). Genes indicated in red are also DE in Fulani +*P.fal* relative to Mossi +*P.fal* (FDR < 0.1). * Indicates location of uncharacterised loci associated with resistance to severe *P.falciparum* malaria. (f) Difference in DNA methylation in selected regions that include cytobands enriched in DE genes (Fulani +*P.fal* relative to Fulani −*P.fal*, adjusted p-valued < 0.05), showing individual probes (black) and binned values (300 probes; red).

DOI: https://doi.org/10.7554/eLife.29156.012

The following source data and figure supplement are available for figure 4:

**Source data 1.** qRT-PCR analysis of selected DE chromatin regulation genes in monocytes of validation cohort of Mossi −*P.fal* (n = 4), Fulani −*P.fal* (n = 10), Mossi +*P.fal* (n = 8) and Fulani +*P.fal* (n = 5) (mean ±SD).
DOI: https://doi.org/10.7554/eLife.29156.014

**Figure supplement 1.** Difference in DNA methylation at between Mossi −*P.fal* (n = 3), Mossi +*P.fal* (n = 3), Fulani −*P.fal* (n = 3), and Fulani +*P.fal* (n = 3).
DOI: https://doi.org/10.7554/eLife.29156.013

inflammasome. Activation of the inflammasome occurs in two stages: first, a 'priming' signal activates expression of both inflammasome components and its target proteins; then, a second 'activation' signal promotes assembly of inflammasome components into an active complex that processes target proteins into their biologically active forms (*He et al., 2016*). Our results showed significantly increased expression of an inflammasome pathway activator (*P2RX7*) in uninfected Fulani compared to Mossi, and then significantly increased levels of target protein activation (IL-1β and IL-18) together with reduced expression of inflammasome pathway components in infected Fulani. This is consistent with reports that activation of the inflammasome results in downregulation of expression of its components. Collectively, this suggests that the Fulani have higher baseline levels of expression of inflammasome pathway components, resulting in stronger inflammasome activation following *P.falciparum* infection.

Recently, a small number of reports have begun to address the role of the inflammasome in the immune response in malaria. The available evidence indicates *P.falciparum* infection can drive inflammasome activation in human patients: In mice, hemozoin or symptomatic *Plasmodium* infection trigger inflammasome formation in erythrocytes and monocytes, respectively (*Ataide et al., 2014*; *Kalantari et al., 2014*); monocytes from *P.vivax* malaria patients contain inflammasome components (*Ataide et al., 2014*); *P.falciparum* and *P.vivax* DNA containing immunocomplexes induce inflammasome assembly and caspase-1 activation in human monocytes in culture (*Hirako et al., 2015*); and primed human monocyte-derived macrophages, incubated with antibody-opsonized *P.falciparum* infected erythrocytes, activate the inflammasome and secrete IL-1β (*Zhou et al., 2012*). Our results showing higher levels of inflammasome activation in *P.falciparum* infected Fulani support these findings, and further, suggest that activation of the inflammasome may be protective and involved in a more effective immune response in malaria.

In regions of hyperendemic malaria, such as where the Fulani and the Mossi included in this study reside, individuals commonly experience multiple *P.falciparum* infections in a transmission season. In the case of other infections, monocytes and other cells of the innate immune system have been reported to develop 'memory', enabling more efficient response to secondary infections (*Netea et al., 2016*). In contrast to innate immune tolerance, this response is typified by increased expression of different gene modules, upregulated proinflammatory cytokine response (*Saeed et al., 2014*), and a shift toward aerobic glycolysis (*Cheng et al., 2014*). This 'trained immunity' is based on epigenetic reprogramming of the innate immune cells (*Netea et al., 2016*; *Glass and Natoli, 2016*).

We hypothesised that the underlying mechanism for the stronger transcriptional response in monocytes from infected Fulani is caused by epigenetic and chromatin differences between Fulani and Mossi. However, the mechanisms mediating the observed transcriptional response in monocytes from infected Fulani were not correlated to changes in DNA methylation. Instead, our results lead us to propose that the transcriptional response in monocytes from infected Fulani is a result of an altered chromatin landscape caused by differential expression of chromatin remodelling and histone modification genes upon infection. Changes in chromatin structures and nuclear architecture, which depend on epigenetic changes as well as on density and spacing of nucleosomes, occur both during differentiation and activation of immune cells (*Schultze, 2017*; *Netea et al., 2016*; *Glass and Natoli,*

2016), and they drive changes in gene expression. The chromatin alterations predicted in Fulani are complex, since differentially expressed genes are classified both as activators and repressors; for example, histone modifying enzymes such as SUV39H1 and SUV420H1, which are responsible for establishing heterochromatin were downregulated, as were those involved in establishing euchromain and activate promoters, such as SET1 and SWI/SNF. In addition, non-coding RNAs contribute to the induction of chromatin states (*Rutenberg-Schoenberg et al., 2016*) and the few DE genes upregulated in the infected Fulani (144/1239) were enriched in non-coding RNA genes. Our conclusion that *P.falciparum* infections induce alteration in the chromatin landscape of monocytes is further supported by the observation that DE genes in Fulani monocytes often cluster in regions. Interestingly, one of these regions, 8q24.3 contains: the *c-Myc* gene; *GSDMD,* encoding inflammasome substrate gasdermin D (*Liang and Liu, 2016*); and a novel, uncharacterised locus of resistance to severe malaria (*Malaria Genomic Epidemiology Network et al., 2015*). Taken together we suggest that genome wide chromatin alterations occur as a result of *P.falciparum* infection in monocytes from Fulani, enabling a heightened transcriptional response.

Cells of the innate immune system, such as monocytes, macrophages and NK cells, have been shown to exhibit a memory alongside the more well studied memory of the adaptive immune system. The innate memory functions, such as 'immune tolerance' and 'trained immunity', can prepare the response to a second stimulation after an earlier first infection. Studies in model systems show that signalling pathways and transcription factors activated by the initial stimulus establish different histone modifications and nucleosome occupancy at specific sites (*Foster et al., 2007*; *Netea et al., 2016*; *Glass and Natoli, 2016*; *Quintin et al., 2014*; *Hoeksema and de Winther, 2016*). These imposed changes cause differences in the transcriptional response to subsequent infections. Trained immunity is characterised by an increased responsiveness of innate immune cells, typified by increased pro-inflammatory cytokine release and antimicrobial activity (*Rizzetto et al., 2016*; *Netea and van der Meer, 2017*) that can persit months after an initial stimulus (*Netea et al., 2016*; *Quintin et al., 2014*). Trained immunity correlates with epigenetic reprogramming at certain sites; ChIP sequencing analysis of monocytes show higher levels of H3K4me1/2 and H3K27Ac at specific gene promoters and enhancers, many of which are involved in pro-inflammatory response, metabolism, and phagocytosis (*Saeed et al., 2014*; *Cheng et al., 2014*; *Novakovic et al., 2016*). Our results show a response that resembles trained immunity in the Fulani. For example, chromatin factors that are involved in H3K4-methylation were significantly differentially expressed. Further, differentially expressed genes reflect typical processes associated with trained immunity, including altered glucose metabolism, lysosome function, and a pro-inflammatory state. We were also able to validate that uninfected Fulani have higher levels of pro-inflammatory cytokines. However, it is difficult to directly compare published data describing the initial and subsequent responses in models of controlled infection, to the responses we observed here in members of a natural population that undergo multiple infections during a malaria transmission season. We do not know when the changes underlying the transcriptional response are established in Fulani individuals – during a recent infection, early in the transmission season, or even earlier in life, during immune development. Nevertheless, our results suggest that in monocytes of uninfected Fulani, chromatin is already set in 'primed' state, driving the observed heightened transcriptional response and pro-inflammatory state. Thus, Fulani display a 'high alert' immune state, enabling a stronger reaction upon *P.falciparum* infection.

## Materials and methods

### Study population and study area

The study sample collection was performed in November 2014. Adolescent men aged 15–24 years were included in the study. The study area was the villages of Barkoundouba Peulh inhabited by Fulani and Barkoumbilen Mossi inhabited by Mossi, located 5 km apart near Ziniaré (Province of Oubritenga) about 35 km North-West of Ouagadougou, the capital city of Burkina Faso. Both communities have been settled in the area for over 50 years. Subjects included in the study were considered permanently resident in the area, as movements have previously been reported to be mostly daily and short range (*Modiano et al., 1996*). The area is shrubby savanna of the Mossi central plateau, approximately 300 m above sea level. The climate is typical of the sudan-sahelian zone, with a

dry season from November to May, and a rainy season from June to October in which malaria transmission is hyperendemic with inoculation rates reaching above one infective bite per person per night (*Modiano et al., 1996*; *Esposito et al., 1988*). Ouagadougou (World Meterological Organization station reference 655030) recorded below average rainfall in October of 2014, suggesting a shorter than average transmission season prior to sample collection. The annual entomological inoculation rates range from 10 to 500 infective bites per individual. The main malaria vectors are two chromosomal forms (Savanna and Mopti) of *Anopheles gambiae*, *An. arabiensis* and *An. funestus* (*Petrarca et al., 1986*). Over 90% of malaria infections are by *P.falciparum* (*Rizzo et al., 2011*).

## Sample collection

Venous whole blood (approximately 18 ml) was collected from study participants, with two additional whole blood spots for diagnosis of *Plasmodium* infection. Peripheral blood mononuclear cells (PBMCs) were purified by Ficoll-Paque density centrifugation according to manufacturer's instruction (GE Healthcare, Uppsala, Sweden). CD14+ monocytes were isolated with EasySep CD14 positive selection kit according to manufacturer's instructions (Stem Cell Technologies, Grenoble, France). CD14+ monocytes and the remaining CD14- cells (lymphocytes) were stored in RNAprotect Cell Reagent (Qiagen, Hilden, Germany) at −80°C (Materials and Methods - Supplement Figure 1). Plasma was obtained following centrifugation of venous whole blood, and stored at −80°C.

## Diagnosis of plasmodium infection

Malaria rapid diagnostic test (RDT) was used to diagnose infected individuals at the time of sample collection. Further diagnosis of *Plasmodium* infection was by microscopic examination of Giemsa-stained blood film, followed by real-time PCR of DNA extracted from whole blood spots for specific *Plasmodium* species (as described by *Shokoples et al., 2009*), with *P.falciparum* positive samples defined as <35 cycles.

## DNA and RNA preparation

Total DNA and RNA were extracted from CD14+ and CD14- cells with AllPrep DNA/RNA Micro Kit according to manufacturer's instructions (Qiagen). Integrity of DNA and RNA was analysed by Agilent Bioanalyser DNA and RNA kits according to manufacturer's instructions (Agilent Technologies, Santa Clara, CA, USA).

## Selection of samples for RNA-sequencing and DNA methylation analysis

Samples selected for RNA-sequencing and DNA methylation were subject to strict criteria: for each sample diagnosis of *P.falciparum* infection was confirmed by PCR analysis; samples with mixed *Plasmodium* infections, G6PD-deficiency or carriers of hemoglobin C were excluded; DNA and RNA quality and integrity was RIN-value >9. According to these criteria, samples from infected and uninfected individuals were included for analysis, as described in *Figure 1—figure supplement 1*. For RNA-sequencing of CD14+ monocytes, four uninfected Mossi and seven uninfected Fulani, and three infected Mossi and three infected Fulani were selected; for RNA-sequencing of CD14-, five uninfected Mossi and five uninfected Fulani, and three infected Mossi and three infected Fulani were selected; for DNA methylation analysis, three uninfected Mossi and three uninfected Fulani, and three infected Mossi and three infected Fulani, exhibiting the least *P.falciparum* DNA in blood by PCR (>39 cycles), were selected. Samples from the same individuals were included across the analyses. The one monocyte sample from infected Fulani was not successfully sequenced, but conversion of the remaining RNA to cDNA and analysis of several highly expressed genes gave similar results as the two sequenced samples (*Supplementary file 2G*). Samples from individuals not included in the RNA-sequencing and DNA methylation analysis formed the cohort used for validation experiments.

## RNA-Sequencing

RNA-sequencing was performed with strand-specific TruSeq libraries on an Illumina HiSeq High Output Mode with paired-end 50 bp reads. Quality control of raw read files was performed with FastQC (0.11.2). Adapters and low quality reads were removed with Trimmomatic (0.32), and the remaining reads were aligned to the Human GRCh38 reference genome with Star (2.4.1 c), with default

parameters. Read counts per exonic region were extracted with the featureCounts (1.5.0) R package using the Homo_sapiens.GRCh38.83.gtf file as reference annotation. MultiQC (0.3.1) was used to aggregate results from multiple samples. Differential expression analysis was performed with the edgeR-Limma (3.24.15) pipeline, using the read count table as input and filtering genes that were lowly expressed. After the comparisons, genes were considered as statistically up- or down-regulated based on Benjamini-Hochberg-corrected p-values with a cutoff of 5%. Analysis of RNA-sequencing reads was performed on resources provided by SNIC through Uppsala Multidisciplinary Center for Advanced Computational Science (UPPMAX), with support by Bioinformatics Infrastructure for Life Sciences (BILS).

## DNA methylation analysis

DNA methylation analysis was performed using the Illumina Infinium HumanMethylation450 Bead-Chip according to manufactures recommendation. IDAT files of Illumina 450K chip were processed by minfi package and further stratified quantile normalized by shinyMethyl package. A total of 164 failed probes, which had been found with detection p-value higher than 0.01 in more than 3 samples of the total, were eliminated from further analysis. Due to only male samples being enrolled in this study, probes that locate on sex chromosomes were included in the methylation analysis. Differential methylation were tested using M-value by linear regression with Limma package both for infected group versus non-infected group (Fulani infected versus non-infected, Mossi infected versus non-infected) and between the two ethnic groups (Fulani non-infected versus Mossi non-infected). Differentially methylated cytosines were defined as fold change $\geq$1.5 and Benjamini-Hochberg adjusted p-value$\leq$0.05.

Methylation data of differentially expressed genes in uninfected versus infected Fulani samples were extracted and genomic annotation of each gene was obtained from ensemble genome build. In total, 18897 450K probes were found covering 940 genes for their full length, and mean value of annotated probes was taken for presenting gene methylation level. Spearman correlation coefficient was computed for average methylation levels of each group for those differentially expressed genes.

## Data availibility

RNA-sequencing and DNA methylation data have been deposited in NCBI's Gene Expression Omnibus and are accessible through GEO Series accession number GSE100563 (https://www.ncbi.nlm.nih.gov/geo/query/acc.cgi?acc=GSE100563).

Quantitative reverse-transcription PCR

cDNA was synthesized from CD14+ monocyte RNA using SuperScript III Reverse Transcriptase Kit (Invitrogen, Carlsbad, CA, USA) with random primers, and qRT-PCR amplification of target sequences was performed in duplicate technical replicates with SYBR Green Chemistry (Kapa Biosystems, Willmington, MA, USA), the calculation of expression were performed with the $2^{-\Delta\Delta Ct}$ method, with PP1A or RPLP1 used for normalization. For primer sequences see (*Table 1*).

## ELISA

Levels of IFNγ, IL-1β, IL-6, IL-10, TGF-β, IL-18, IL-1Ra Il-1RII, and IL-18BP were measured by ELISA according to manufacturer's instructions (Mabtech, Nacka, Sweden. Cat. # 3420-1A-6 (IFNγ), # 3415-1A-6 (IL-1β), # 3460-1A-6 (IL-6), # 2430-1A-6 (IL-10), # 3550-1A-6 (TGFβ), using plasma samples diluted 1:2 in ELISA diluent (Mabtech Cat. # 3652-D); MBL International, Woburn, MA, USA. Cat # 7620 (IL-18); R and D Systems, Minneapolis, MN, USA. Cat # DRA00B (Il-1Ra), # DR1B00 (IL-1RII), # DBP180 (IL-18BPa). Levels of IgG against *P.falciparum* schizont extract antigens of the 3D7 strain were measured by ELISA as previously described(*Bolad et al., 2005*).

## Statistics

Statistical analysis was performed with two-tailed Fisher's exact test (*P. falciparum* infection rates, G6PD status), two-tailed Mann-Whitney non-parametric U test (age, temperature, Hb *levels*, *P. falciparum* density, ELISAs)

**Table 1.** Primer sequences used for QRT-PCR analysis

| Gene | F | R |
| --- | --- | --- |
| ADGRG3 | CGAAGGGCCAAGAAACACCT | CGTAGTTTAGCCAGTATCTCTGC |
| ALDOA | ATGCCCTACCAATATCCAGCA | GCTCCCAGTGGACTCATCTG |
| ARID3A | AGCTGCAGCCGCCTGACCAC | TGTTGGGAGCAGAGGTTGGC |
| BRG1 (SMARCA4) | AGGCAAAATCCAGAAGCTGA | CGCTTGTCCTTCTTCTGGTC |
| CATSPER2 | ATGGCCGCTTACCAACAAGAA | TGCAAATGCTCAATGAGAGAGAA |
| CPVL | TGGAAGGTGATTGTTTCGCTG | GTCTCCCTTAGGTGGCATGGA |
| CXXC1 | GCAAACCGGACATCAACTGC | GCACTCCCGACAGTACCAC |
| DACH1 | GGGGCTTGCATACGGTCTAC | CGAACTTGTTCCACATTGCACA |
| DUSP8 | CGAACTTGTTCCACATTGCACA | CGAACTTGTTCCACATTGCACA |
| EPBH1 | GGCTGCGATGGAAGAAACG | CTGGTTGGGCTCGAAGACATT |
| FAM26F | TGTCACCCGATGCCTATCTC | TGGCCCTTCGGATTGAAAGTA |
| FES | GGCCGAGCTTCGTCTACTG | GTCCTGCATACTCCCTGTCAC |
| FURIN | CCTGGTTGCTATGGGTGGTAG | AAGTGGTAATAGTCCCCGAAGA |
| GAA | CATCCTACTCCATGATTTCCTGC | AGCTGGGTGAGTCTCCTCC |
| GAPDH | TGCACCACCAACTGCTTAGC | GGCATGGACTGTGGTCATGAG |
| IRF3 | IRF3 Set #VHPS-4629 purchased from Realtimeprimers.com | |
| IRF5 | IRF5 Set #6 purchased from Realtimeprimers.com | |
| KDM4B | KDM4B Set #6 purchased from Realtimeprimers.com | |
| KDM6B | KDM6B Set #1 purchased from Realtimeprimers.com | |
| NFKBIE | TCTGGCATTGAGTCTCTGCG | AGGAGCCATAGGTGGAATCAG |
| OSM | CACAGACTGGCCGACTTAGAG | AGTCCTCGATGTTCAGCCCA |
| P2R × 7 | CTCCCATCTCAACTCCCTGA | TCCTGGTAGAGCAGGAGGAA |
| PHC2 | AGGGAACGGAAACTCTGCCT | TCGATAACATGCGTCAGGATTTG |
| PP1A | AGACAAGGTCCCAAAGAC | ACCACCCTGACACATAAA |
| PPP4C | AAGGTTCGCTATCCTGATCGC | AGCCATAGACCTGCGTGATCT |
| RFXANK | GTGACAACCTCGTCAACAAGC | CGAACGGTCTCAATCTCTCCAA |
| RUVBL2 | RUVBL2 Set #1 purchased from Realtimeprimers.com | |
| SETD1A | SETD1A Set #1 purchased from Realtimeprimers.com | |
| SIRT7 | GTGGACACTGCTTCAGAAG | CACAGTTCTGAGACACCACA |
| SQSTM1 | GCACCCCAATGTGATCTGC | CGCTACACAAGTCGTAGTCTGG |
| STRA13 | CCTCCTGGCCACATTCCTG | GATTTATTGATGTTGCTTTGTGAGAA |
| SUV39H1 | SUC39H1 Set #2 purchased from Realtimeprimers.com | |
| TOLLIP | AGA ATC CCC GCT GGA ATA AG | GCG TAG GAC ATG ACG AGG TT |
| TRAF7 | TCTGCGCTCCACATTCTCAC | ACCGCGATGTTGTTCACCA |

DOI: https://doi.org/10.7554/eLife.29156.015

## Ethics approval

The study protocol and the informed consent were approved by the Institutional Review Board, The Technical Committee of the Centre National de Lutte contre le Paludisme of the Ministry of Health of Burkina Faso (2014/065/MS/SG/CNRFP/CIB). The study was conducted in compliance with International Conference on Harmonization's Good Clinical Practice principles, the Declaration of Helsinki, and the regulatory requirements of Burkina Faso.

## Acknowledgements

The authors wish to thank all the study participants for their kind contributions and willingness to take part in this study, and all the laboratory staff and field workers for their outstanding efforts towards the success of this study. The authors acknowledge support from Science for Life Laboratory, the Knut and Alice Wallenberg Foundation, the National Genomics Infrastructure funded by the Swedish Research Council, and the Uppsala Multidisciplinary Center for Advanced Computational Science for assistance with parallel sequencing and access to the UPPMAX computational infrastructure. The authors are grateful to Thomas Källman and Olga Dethlefsen of Bioinformatics Infastructure for Life Sciences. JEQ is supported by a scholarship from The Sven and Lilly Lawskis Fund for Scientific Research.

## Additional information

### Funding

| Funder | Grant reference number | Author |
| --- | --- | --- |
| Stockholm University | SciLife Pilot Grant | Mary A O'Connell<br>Marita Troye-Blomberg<br>Ann-Kristin Östlund Farrants |
| BioMalPar European Network of Excellence | LSHP-CT-2004-503578 | Ioana Bujila<br>Marita Troye-Blomberg |
| Seventh Framework Programme | FP7/2007-2013 N 242095 | Marita Troye-Blomberg |
| Sven and Lilly Lawskis Fund | | Jaclyn E Quin |

The funders had no role in study design, data collection and interpretation, or the decision to submit the work for publication.

### Author contributions

Jaclyn E Quin, Conceptualization, Formal analysis, Validation, Investigation, Visualization, Methodology, Writing—original draft, Writing—review and editing; Ioana Bujila, Andreas Lennartsson, Conceptualization, Formal analysis, Investigation, Methodology, Writing—review and editing; Mariama Chérif, Conceptualization, Validation, Investigation, Methodology, Writing—review and editing; Guillaume S Sanou, Investigation, Project administration, Writing—review and editing; Ying Qu, Formal analysis, Visualization, Writing—review and editing; Manijeh Vafa Homann, Investigation, Methodology, Writing—review and editing; Anna Rolicka, Investigation, Writing—review and editing; Sodiomon B Sirima, Project administration, As director of Centre National de Recherche et de Formation sur le Paludisme, he was instrumental in the co-ordination of ethical compliance and material aquisition in Burkina Faso. Without his support, the study could not have been conducted; Mary A O'Connell, Marita Troye-Blomberg, Conceptualization, Investigation, Methodology, Writing—review and editing; Issa Nebie, Conceptualization, Investigation, Project administration, Writing—review and editing; Ann-Kristin Östlund Farrants, Conceptualization, Formal analysis, Supervision, Funding acquisition, Investigation, Methodology, Writing—original draft, Writing—review and editing

### Author ORCIDs

Jaclyn E Quin http://orcid.org/0000-0001-7757-7295
Guillaume S Sanou http://orcid.org/0000-0003-0536-616X
Ann-Kristin Östlund Farrants http://orcid.org/0000-0001-9225-3264

### Ethics

Human subjects: The study protocol and the informed consent were approved by the Institutional Review Board, The Technical Committee of the Centre National de Lutte contre le Paludisme of the Ministry of Health of Burkina Faso (2014/065/MS/SG/CNRFP/CIB). The study was conducted in compliance with International Conference on Harmonization's Good Clinical Practice principles, the Declaration of Helsinki, and the regulatory requirements of Burkina Faso.

**Decision letter and Author response**
Decision letter https://doi.org/10.7554/eLife.29156.021
Author response https://doi.org/10.7554/eLife.29156.022

## Additional files

**Supplementary files**

• Supplementary file 1. (A) Significantly differentially DNA methylated genes in monocytes of Fulani -*P.fal* relative to Mossi –*P.fal*. (B) Significantly differentially DNA methylated genes in monocytes of Fulani +*P.fal* relative to Fulani –*P.fal*.
DOI: https://doi.org/10.7554/eLife.29156.016

• Supplementary file 2. (A) List of all significantly DE genes in monocytes for all comparisons (adjusted p-value<0.05, shown in red). Where gene symbol is listed in red, gene is not significantly DE in Fulani +*P.fal* relative to Fulani –*P.fal*. (B) List of significantly DE genes in monocytes of Fulani +*P.fal* relative to Fulani –*P.fal* (adjusted p-value<0.05). (C) List of significantly DE genes in monocytes of Mossi +*P.fal* relative to Mossi –*P.fal* (adjusted p-value<0.05). (D) List of significantly DE genes in monocytes of Fulani +*P.fal* relative to Mossi +*P.fal* (adjusted p-value<0.05). (E) List of inflammasome related significantly DE genes in monocytes of Fulani + *P.fal* relative to Fulani –*P.fal* (adjusted p-value<0.05). (F) List of transcription factor binding sites enriched at significantly DE genes in monocytes of Fulani +*P.fal* relative to Fulani –*P.fal*. (G) qRT-PCR analysis of selected DE genes in Fulani cohort for RNA-sequencing analysis. Fulani –*P.fal* (n=7) and Fulani +*P.fal* (n=3) (mean ± SD).
DOI: https://doi.org/10.7554/eLife.29156.017

• Transparent reporting form
DOI: https://doi.org/10.7554/eLife.29156.018

**Major datasets**

The following dataset was generated:

| Author(s) | Year | Dataset title | Dataset URL | Database, license, and accessibility information |
|---|---|---|---|---|
| Jaclyn E Quin, Ioana Bujila, Ying Qu, Andreas Lennartsson, Ann-Kristin Östlund Farrants | 2017 | Major transcriptional changes observed in the Fulani; an ethnic group less susceptible to malaria | https://www.ncbi.nlm.nih.gov/geo/query/acc.cgi?acc=GSE100563 | Publicly available at the NCBI Gene Expression Omnibus (accession no. GSE100563) |

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
