## [Decision Letter]

Thank you for submitting your article "Major transcriptional changes observed in the Fulani; an ethnic group less susceptible to malaria" for consideration by *eLife*. Your article has been favorably evaluated by Tadatsugu Taniguchi as the Senior Editor, Urszula Krzych as the Reviewing Editor, and three reviewers. The reviewers have opted to remain anonymous.

The reviewers have discussed the reviews with one another and the Reviewing Editor has drafted this decision to help you prepare a revised submission.

Summary:

It is well established that Fulani and Mossi exhibit different infection rates to *P. falciparum* malaria. Several mechanisms have been proposed to account for these differences, however, understanding causes of this differing susceptibility to malaria remains incomplete. In this study the authors have undertaken a pilot study to examine global DNA methylation, which appears not to differ between these sympatric ethic groups, and transcriptomes expression in CD14+ and CD14- peripheral mononuclear cell populations. According to the results presented here, significant differences do appear to exist in the expression of genes involved in innate immune responses. This observation supports previous reports concerning elevated pro-inflammatory responses amongst the Fulani group. The question about the mechanisms responsible for the differences in susceptibility to *P. falciparum* is important towards reaching further understanding of the host-parasite relationship, which inevitably would provide a more targeted support towards a design of effective approaches to eradicate malaria.

The reviewers expressed positive views towards this study, however, they also indicated some concerns that would need to be addressed before the final acceptance of the manuscript.

Essential revisions:

1) Fold-changes and significance levels need to be reported in the supplementary material for a common set of genes across all contrasts (i.e., the union of the 1239 Fulani infected vs. uninfected with those from all other comparisons).

2) With a modest sample size, it would be valuable to include a heatmap that summarizes changes at the sample level, and includes all comparisons in the same plot. Some of the contrasts have an N=6, so it would be helpful to see how much individual samples may be driving results. The authors might consider such a plot for the supplementary material.

3) It is unclear from the Materials and methods section what statistical test was used to derive the RNA-seq p-values. The authors mention that cell populations from the same patients were used. Was a paired analysis performed?

4) Some sections need further discussion or clarification. For example, chromatin remodeling genes are higher in Fulani uninfected samples "like immune genes above", which clearly implies an innate and/or trained state of heightened sensitivity. How does this rationale apply to chromatin remodeling? The authors later mention the connection with trained immunity, but if indeed it makes sense for a cell to maintain higher levels of chromatin remodeling machinery, how does this relate with their interpretation that stable epigenetics (as opposed to DNA methylation) is the underlying mechanism? Does the description of DE genes clustering in genomic regions refer to analysis from the current work?

---

## [Author Response]

Essential revisions:1) Fold-changes and significance levels need to be reported in the supplementary material for a common set of genes across all contrasts (i.e., the union of the 1239 Fulani infected vs. uninfected with those from all other comparisons).

To address the reviewers’ request for fold-change and significance levels across a common set of genes set for all contrasts, we have now included a table that incorporates all significantly DE genes for all comparisons, with fold-change and significance levels (log2FC, p-values, and adjusted p-values) shown for each gene in each comparison. This is included in Supplementary file 2 (Supplementary file 2 – All significantly DE genes). This now enables the reader to easily identify whether any significantly DE genes from one comparison are significantly DE in any other comparison.

2) With a modest sample size, it would be valuable to include a heatmap that summarizes changes at the sample level, and includes all comparisons in the same plot. Some of the contrasts have an N=6, so it would be helpful to see how much individual samples may be driving results. The authors might consider such a plot for the supplementary material.

To summarise the relationships between all the samples included in the RNA-sequencing analysis, and the degree of differences between each individual sample, we have now included a heat map as the reviewers suggested (Figure 2—figure supplement 1), as well as Multidimensional scaling (Figure 2—figure supplement 2). In both analysis, *P. falciparum* infected Fulani cluster together, and are most clearly different from other populations, as is reflected in our RNA-sequencing analysis of significantly DE genes. This indicates that it is no one single individual driving our RNA-sequencing results.

In the heat map (Figure 2—figure supplement 1), one uninfected Fulani individual clusters with the *P. falciparum* infected Fulani. However, in Multidimensional scaling this uninfected Fulani individual now clusters separately (corresponding to the Fulani -*P.fal* sample farthest to the right on the X-axis). We have also previously performed RNA-sequencing analysis for only those individuals included in the DNA-methylation analysis, which excluded this individual (Mossi *–P.fal* (n=3), Mossi *+P.fal* (n=3), Fulani *–P.fal* (n=3), and Fulani *+P.fal* (n=2)). Over 70% of significantly DE genes in Fulani +*P.fal* v Fulani –*P.fal* are also significantly DE in this more limited comparison. Therefore, we are confident this individual is not unduly impacting our RNA-sequencing analysis.

3) It is unclear from the Materials and methods section what statistical test was used to derive the RNA-seq p-values. The authors mention that cell populations from the same patients were used. Was a paired analysis performed?

We were remiss to not initially include the details of the statistical analysis used to calculate the RNA-sequencing p-values, and thank the reviewers for pointing this out. Differential expression analysis was performed with the edgeR-Limma (3.24.15) pipeline. We used the function calcNormFactors from the package edgeR to normalize the libraries and estimate library size. After this the read counts were converted to log counts per million using the voom function from the limma package. This function also estimates weights for individual genes and libraries that are used in modelling the gene expression using the lmFit function from the limma package. We used a simple model to identify differentially expressed genes in CD14+(monocytes) and CD14- (lymphocytes) independently. Genes were considered as statistically up- or down-regulated based on Benjamini-Hochberg-corrected p-values with a cutoff of 0.05. This information is now included in Materials and methods, see subsection “RNA-Sequencing”. Paired analysis was not performed, as we did not include any comparisons of expression between CD14+ (monocyte) and CD14- (lymphocytes), which were from the same individual.

4) Some sections need further discussion or clarification.

The reviewers suggested that some sections in the text needed further discussion or clarification. We have now expanded these sections and hope that we have been able to address the questions to make our reasoning more clear. The relevant sections are outlined here.

For example, chromatin remodeling genes are higher in Fulani uninfected samples "like immune genes above", which clearly implies an innate and/or trained state of heightened sensitivity. How does this rationale apply to chromatin remodeling?

We find that genes that have increased expression in response to *P. falciparum* infection in the control group, Mossi, are already expressed at higher levels in uninfected Fulani when analysed by qPCR in the validation cohort. We think that this is an interesting observation, and as these epigenetic and chromatin factors are also part of this pattern, we speculate that they may be involved in mediating the monocyte response to *P. falciparum* and the heightened state of expression in uninfected Fulani. The initially higher gene expression in Fulani established by a changed chromatin landscape could conceivably be formed early in life or by more recent infections. We have now re-written this section and clarified our reasoning, see changes in the last paragraph of the Results section.

The authors later mention the connection with trained immunity, but if indeed it makes sense for a cell to maintain higher levels of chromatin remodeling machinery, how does this relate with their interpretation that stable epigenetics (as opposed to DNA methylation) is the underlying mechanism?

Chromatin changes are involved in the differentiation and activation of immune cells, and these changes are usually regulated by specific transcription factors that recruit different DNA methylation enzymes, histone modifying enzymes and ATP-dependent chromatin remodelling complexes. We did not find any differential DNA methylation pattern, but instead a large number of histone modifying and chromatin remodelling genes were differentially expressed in monocytes from infected Fulani. This lead to the proposal that the underlying stable chromatin state in these cells are different, however it is impossible from our material to predict in which way and when the chromatin structures are established. It would be interesting to investigate whether these more stable chromatin states make chromatin remodelling genes more prone to be expressed in Fulani upon infection, thereby allowing for changed accessibility of the regulatory regions of other genes. The material collected in the present study was not sufficient to approach this aspect by ATAC-seq or Mnase seq.

Since many of the most significantly DE genes in monocytes from infected Fulani were involved in processes that were also identified by RNA-sequencing in a model of trained immunity, we suggest that the monocytes in Fulani exist in a state that are easily activated in a way resembling trained immunity. The DE genes are involved in processes associated with trained immunity, i.e. genes involved in glucose metabolism, lipid metabolism, phagocytosis and lysosome function, and inflammatory mediators. Activation of trained immunity is linked to specific histone modification at regulatory regions, mainly by histone H3K4me and H3K27Ac. Chromatin modifying genes associated with such modifications were also significantly DE in infected Fulani monocytes. The stronger transcriptional response suggests that the chromatin was already set in a more “high alert” state priming for the response to infection. However, the controlled infections typical in studies investigating the responses in trained immunity differ from the natural populations with multiple infections investigated here, therefore we only suggest similarities in responses and say that the monocyte from Fulani are in a high alert state.

Clarification of this question is the most significant change we have made to the text. See section starting: Results, last paragraph; Discussion, fourth and fifth paragraphs.

Does the description of DE genes clustering in genomic regions refer to analysis from the current work?

In short, yes. We have now clarified this in the last paragraph of the Results section.